# Adjustments in Food Choices and Physical Activity during Lockdown by Flemish Adults

**DOI:** 10.3390/nu13113794

**Published:** 2021-10-26

**Authors:** Evelien Mertens, Peter Deriemaeker, Katrien Van Beneden

**Affiliations:** 1Department of Health Care, Design and Technology, Nutrition and Dietetics Program, Erasmushogeschool Brussel, 1090 Brussels, Belgium; peter.deriemaeker@ehb.be (P.D.); katrien.van.beneden@ehb.be (K.V.B.); 2Department of Movement and Sport Sciences, Faculty of Physical Education and Physiotherapy, Vrije Universiteit Brussel, 1050 Brussels, Belgium

**Keywords:** food choices, nutrition, dietary intake, physical activity, sedentary behaviour, lockdown, adults, COVID-19

## Abstract

Background: On Wednesday 18/03/2020 Belgium was placed in lockdown in order to curb the spread of COVID-19. Lockdown can lead to loneliness, boredom, anger, anxiety and depression, which in turn have an influence on food choices and physical activity (PA). This study aims to map the adjustments in food choices and PA by Flemish adults during lockdown. Methods: Chi square tests were performed to investigate the relationship between adjustments in food choices, PA and demographic variables. Results: A total of 1.129 respondents filled in the online questionnaire, aged between 18 and 81 years. The healthiest food choices were made by respondents living alone during lockdown, whilst people cohabiting with others increased their PA significantly. Moreover, the dietary adjustments of adults living with children evolved more favourably to healthier choices then those cohousing with other adults. However, respondents living with other adults showed a more favourable pattern regarding adjustments in PA. The strongest increase in sedentary behaviour was observed in students. Conclusions: This study shows the impact of lockdown on both PA and food choices, where healthier adjustments were observed in PA and respondents were prone to consume unhealthier food.

## 1. Introduction

At the end of 2019, the novel coronavirus SARS-CoV-2 spread rapidly from the Chinese city of Wuhan to the rest of the world, causing an atypical pneumonia named COVID-19 [1,2]. In 2020, specifically on March 18, the Belgian government decided to impose a nationwide lockdown in order to curb the spread of COVID-19: schools from kindergarten to higher education, bars, sport clubs and non-essential shops were closed. As sport clubs were closed during lockdown the Belgian population was restricted to exercise at home or outside with a maximum of one extra person. Food acquisition was restricted to supermarkets, grocery stores and (restaurant) delivery/take-away services. To further reduce the transmission of COVID-19, social contacts and interactions with people from different households were limited by cancelling social events andstrongly encouraging employ, sport activities and by strongly encouraging employees to telework from home. All the above-mentioned directives initially led to more staying at home and stockpiling non-perishable food and toilet paper [3].

The outbreak of a pandemic such as COVID-19 can trigger mental health problems such as anxiety, stress and depression because of uncertainty and panic [4]. A recent study showed that the COVID-19 outbreak increased anxiety, depression and alcohol consumption in the Chinese population. More research should be done, however it is likely that these findings of decreased mental well-being can be confirmed in other populations as well [5]. Food choices are strongly influenced by various environmental and economic issues, as well as emotions (e.g., boredom, anxiety, depression, …) and social aspects [6,7,8].

When looking at the impact of stress on physical activity (PA), the majority of studies show that stress has a negative impact on the amount of PA, especially in the elderly and novice athletes [9]. In contrast, other studies show an increase in PA, especially in those who are already regularly physically active prior to a stressful period [9]. However an increase in sedentary behaviour is also reported in the literature in times of acute psychological stress [9].

Given the fact that lockdown is an exceptional situation without precedent, only a few studies can be found investigating both food choices and PA parameters during such a period. As nutritional habits and food choices show country-specific and regional variations, investigating changes in food choices and PA parameters during lockdown according to specific groups in the population may provide essential information for policy makers [10,11].

Di Renzo et al. (2020) reported that Italian subjects between the ages of 18 and 30 showed a higher adherence to the Mediterranean Diet during lockdown compared to younger and older age groups. A trend towards purchasing more farmer or organic fruits and vegetables was seen in 15% of the behaviour of respondents [12]. According to Cancello et al. (2020) 34% improved their diet quality [13]. A study investigating the eating habits in Spain during lockdown outlined healthier food choices compared to their habits before lockdown, as many subjects decreased the intake of fried foods, snacks, fast-food, red meat, pastries and sweet beverages, and increased foods such as olive oil, vegetables, fruits or legumes [14]. Similar results were found in other countries, ranging from the United Arab Emirates, Middle East and North Africa and Lithuania, where fast-food consumption generally decreased during lockdown [15,16,17,18]. Furthermore, several studies reported a decrease in the frequency of ordered food during lockdown [18,19,20]. However, the study of Zajacova et al. (2020) showed an increase of fast-food consumption during early stages of the pandemic in Canada [21]. In contrast to Rodriquez-Perez et al. (2020), Batlle-Bayer et al. (2020) reported a COVID diet in Spain with larger energy intake and lower nutritional quality. An increase of 30–35% on Global Warming Potential, Blue Water Footprint and Land Use in association with dietary changes due to lockdown was shown [22]. Furthermore, a Polish study showed that 28% of the participants adhered to a healthier dietary pattern. Subjects aged over 40 years, those living with children, unemployed people and those not consuming homemade meals were described to have unhealthier food habits [23]. Another Polish study showed that eggs, potatoes, sweets, canned meat and alcohol were consumed considerably more common, whilst bakery products, red meat, fast-food, instant soups, sweet beverages and energy drinks were consumed less frequently [20]. In other studies an increase in snacking was also reported during lockdown [17,18,24,25,26,27,28]. A Chilean study concluded that an increase in body weight during lockdown was positively associated with the consumption of fried foods 3 times or more per week, low water consumption and sedentary time of 6 h or more per day. An inverse association was found with fish consumption, active breaks and being physically active 4 times or more per week [29]. A French study reported more addiction-related habits during lockdown, specifically an increase of caloric/salty food intake, screen time and alcohol use. Associated with an increase in caloric/salty food intake were female gender, aged less than 29 years, having a partner, being in lockdown in a more confined space and being alone during lockdown [30].

The adjustments in PA during lockdown were investigated in different populations and age groups [31]. Di Renzo et al. (2020) reported that body weight training was increased in respondents (aged 12 years and older) and training frequency was increased among those Italians already taking part in sports [12]. Several studies were performed describing PA in different Italian subpopulations. For instance, Gallè et al. (2020) and Luciano et al. (2020) found a decrease in PA and an increase in sedentary behaviour among students [32,33,34]. In contrast Tornaghi et al. (2020) showed that inactive or moderately active students did not alter their PA level, whereas high active ones increased it [35]. Cancello et al. (2020) found a reduction in PA in the majority of the pre-lockdown physically active subjects [13]. According to Di Corrado et al. (2020), about half of the participants maintained their training habits [36]. A decrease in PA was found in all age groups according to Maugeri et al. (2020) [37]. Moreover, Gornicka et al. (2020) showed that about 40% of Polish subjects decreased PA, whilst about a half of the participants increased screen time [23], which suggests that depending on the population the effect of lockdown on PA is omnifarious.

The abovementioned studies provide a general overview regarding dietary or PA changes due to lockdown restrictions in several countries, but only few studies analyzed differences in changes between groups of people. Therefore the aim in the present study is to map in detail the adjustments in different food groups and PA in Flemish adults during lockdown. Furthermore, the relationship was examined between these adjustments in food choices and PA with demographic variables such as gender, area of residence, family situation and work situation. The hypothesis is that in general respondents would prefer more fresh foods, less fast-food (including less take-away), would increase their snacking habits and would change the type of PA in more unstructured/unorganized types of PA.

## 2. Materials and Methods

The adjustments in different food groups and PA were investigated using an online questionnaire. The answer options of the questionnaire were drawn up on a 5-point scale (‘I consume/do this now’: (1) much less than before the lockdown, (2) less than before the lockdown, (3) the same as before the lockdown, (4) more than before the lockdown, (5) much more than before the lockdown). When a food was never part of the diet (for example: meat in a vegetarian dietary pattern), they were asked to indicate score 3 (= the same as before the lockdown). This was also asked when some type of PA was never part of the respondent’s exercise or sports schedule. Table 1 frames the items which were questioned. All questions concerning food choices and PA choices were only surveyed once each. In addition to this information, some general data were collected such as gender, age, area of residence, family situation and work situation. The questionnaire was distributed online using Limesurvey 2.0 between 9 April 2020 and 22 April 2020. Respondents were recruited by social media and email. The minimum age for participation was 18 years. The power analysis showed that a minimum of 226 participants was required to achieve a significant outcome (Effect size w: 0.1865710, α err prob: 0.05, Power: 0.80) [38]. The study was approved by the ethics committee of UZ Brussel (Protocol EMLS2020–B1432020000102).

The statistical analyses were performed using SPSS 26.0 (SPSS Inc., Chicago, IL, USA). Chi square tests were performed to investigate the relationship between adjustments in food choices and types of PA with variables such as gender, age, area of residence, home situation, family situation and work situation. *p* < 0.05 was considered significant. Cut-off values of the most stable and least stable parameters were decided taking into account the lowest and highest percentages of changes in all food groups and types of PA.

## 3. Results

### 3.1. Study Demographics

A total of 1.129 respondents finished the online questionnaire. Table 2 presents the descriptive data of the respondents. The age of the respondents was 34.9 ± 14.3 years (mean ± standard deviation (SD)) with a minimum age of 18.0 years and a maximum age of 81.0 years. The majority (77.1%) of the respondents were women. Most respondents (40.0%) lived in suburban area, followed by rural (31.2% of the respondents) and urban (28.8% of the respondents). 85.6% indicated that they did not live alone, but lived together with others. Of those 85.6% respondents who lived with others, 54.3% lived with other adults (partner, parents, friends, …) and 31.3% lived with children (and possibly a partner). 55.1% of respondents in this study belong to the active population (= employed respondents), 34.5% were students, while 10.4% of the respondents belong to the inactive group (= unemployed people/job seekers, people who receive sickness benefits, people in (pre-)retirement).

### 3.2. Overall Adjustments in Nutritional and PA Parameters during Lockdown

For certain food groups and types of PA, more than 80% of the respondents gave a score of 3 (= “I consume/do this now the same as before the lockdown”). The left column of Table 3 shows the nutritional and PA parameters that fluctuated the least in both directions (this means both “I consume/do this now (much) less than before the lockdown” and “I consume/do this now (much) more than before the lockdown”) and thus remained generally very stable. The nutritional and PA parameters which remained generally very stable were: margarine to bake (84.4% of the respondents consumed this the same as before the lockdown), margarine to spread (84.1%), oils (83.4%), red meat (83.1%), rice (82.3%), white meat (81.9%), cheese spread (81.2%), number of main meals (81.2%), processed meat products (80.3%) and canned seafood/seafood in a jar (80.3%). It should be noted here that respondents who never consumed certain foods or types of PA also indicated a score of 3 (= “I consume/do this now the same as before the lockdown”).

For certain food groups and types of PA, less than 50% of the respondents gave a score of 3 (= “I consume/do this now the same as before the lockdown”). The right-hand column of Table 3 shows the parameters regarding food choices and PA that fluctuated the most in both directions and were therefore generally very unstable. The nutritional and PA parameters which were generally very unstable were: number of snacks (45.5% of the respondents consumed this the same as before the lockdown), takeaway/delivery: (local) restaurant/brasserie/bistro (46.8%), chocolate (44.6%), cookies, cake and/or pie (48.0%), water (including sparkling water and flavoured water) (49.6%), walking (19.7%), cycling (41.3%) and sedentary behaviour (33.0%).

Table 4 illustrates the nutritional and PA parameters that generally decreased and increased the most. The strongest fallers are reported in the right column of Table 4. The percentages shown in Table 4 indicate how many respondents consumed/did this much less or less than before the lockdown. The strongest decreases were seen in takeaway/delivery with 46.0% of the respondents indicating they consumed (much) less takeaway/delivery: international food, 44.4% reported consuming (much) less takeaway/delivery: French fries, 43.4% answered they ordered (much) less takeaway/delivery: (local) restaurant/brasserie/bistro and 42.8% takeaway/delivery: pizza. Meal boxes were also ordered much less to less, but the decrease was lower (33.4%) compared to the other takeaway/delivery services. The strongest risers are reported in the left column of Table 4. The percentages shown in the table represent how many respondents indicated that they consume/do this more or much more than before the lockdown. The strongest risers were 36.5% of the respondents indicated consuming (much) more cookies, cake and/or pie, 37.1% increased their coffee and/or tea intake, 37.9% reported cycling (much) more, 39.2% ate (much) more fresh vegetables, 40.3% ate (much) more eggs, 40.3% increased their chocolate intake, 42.4% enhanced their fresh fruit consumption, 44.8% drank (much) more water (including sparkling water and flavoured water), 55.3% of the respondents increased their sedentary behaviour and 62.7% walked (much) more.

### 3.3. Changes in Nutritional and PA Parameters According to Age (18–34 Year Old versus 35–50 Year Old versus 51–81 Year old) and Gender

Figure 1 provides a schematic overview of the results about the relationship between adjustments in food choices and PA with demographic variables. The colors indicate positive (= green), slightly negative (= orange) or strongly negative (= red) changes or adjustments in food choices and PA according to health. The signs demonstrate which groups are doing better or less well than the others, taking into account the health perspective.

As shown in Appendix A the youngest age group consumed significantly more fresh fruit (χ^2^ = 40,630, *p* < 0.001), pasta (χ^2^ = 25,588, *p* = 0.001), cookies, cake and/or pie (χ^2^ = 27,546, *p* = 0.001), and plant-based dairy (χ^2^ = 46,268, *p* < 0.001), as well as having a strong increase in sedentary behaviour was seen (χ^2^ = 84,740, *p* < 0.001). Moreover, this age category showed a strong decline in beer (χ^2^ = 41,515, *p* < 0.001) and liquor (χ^2^ = 39,748, *p* < 0.001) consumption compared to the other age groups.

In addition, the middle age category consumed significantly more chocolate (χ^2^ = 22,483, *p* = 0.004), crisps (χ^2^ = 27,620, *p* = 0.001), and sweet spreads (χ^2^ = 35,375, *p* < 0.001), while compared to the other age groups a decrease was seen, as the middle age group ate more processed meat products (χ^2^ = 30,035, *p* < 0.001) and drank more wine (χ^2^ = 31,489, *p* < 0.001) during lockdown.

When looking at the oldest age group, a strong increase in milk and dairy consumption (χ^2^ = 15,908, *p* = 0.044) and a strong decrease in consumption of crisps (χ^2^ = 27,620, *p* = 0.001) and soft drinks (χ^2^ = 46,862, *p* < 0.001) was seen. In the oldest age category, a significant reduction in PA was seen for jogging/running (χ^2^ = 86,096, *p* < 0.001) and strength training with own body weight (χ^2^ = 136,367, *p* < 0.001).

Women showed a stronger decrease in beer (χ^2^ = 55,516, *p* < 0.001) consumption and a significant increase in chocolate (χ^2^ = 13,223, *p* = 0.01), cookies, cake and/or pie (χ^2^ = 15,114, *p* = 0.004) and coffee/tea (χ^2^ = 17,495, *p* = 0.002) consumption than men.

The other nutritional and PA parameters were not statistically significant (*p* > 0.05).

### 3.4. Changes in Nutritional and PA Parameters according to Area of Residence (Urban Area versus Suburban Area versus Countryside)

People living in urban areas showed the strongest increase in cookies, cake and/or pie (χ^2^ = 17,912, *p* = 0.022) and sedentary behaviour (χ^2^ = 45,841, *p* < 0.001). The consumption of sauces saw a sharp increase from repondents from an urban area, while people from suburban areas generally reported little adjustments and people from rural areas even indicated a decrease (χ^2^ = 21,279, *p* = 0.006). The strongest increase in the PA parameter walking was seen in people living in rural areas, followed by people from suburban and urban areas (χ^2^ = 49,740, *p* < 0.001). A decrease in cycling was seen in people from urban areas, while there was a sharp increase in other areas (χ^2^ = 66,686, *p* < 0.001).

The other nutritional and PA parameters were not statistically significant (*p* > 0.05).

### 3.5. Changes in Nutritional and PA Parameters according to Home Situation (Living Alone versus Cohabiting)

People living alone showed stronger declines in consumption of takeaway/delivery French fries (χ^2^ = 11,598, *p* = 0.021) and international food (χ^2^ = 10,432, *p* = 0.034), fried potato preparations (χ^2^ = 21,352, *p* < 0.001), red meat (χ^2^ = 23,178, *p* < 0.001), processed meat products (as opposed to an increase in cohabitants) (χ^2^ = 12,754, *p* = 0.013), milk and dairy (χ^2^ = 36,135, *p* < 0.001), fresh cheese (χ^2^ = 11,729, *p* = 0.019) (as opposed to an increase in cohabitants), soft cheese (χ^2^ = 12,785, *p* = 0.012) (as opposed to an increase in cohabitants), margarine for baking (χ^2^ = 17,744, *p* = 0.001) (as opposed to an increase in cohabitants), ice cream and/or sorbet (χ^2^ = 12,191, *p* = 0.016) (as opposed to an increase in cohabitants), sweet spreads (χ^2^ = 13,783, *p* = 0.008) (as opposed to hardly any adjustments in cohabitants), sauces (χ^2^ = 24,800, *p* < 0.001) (as opposed to an increase in cohabitants), beer (χ^2^ = 16,058, *p* = 0.003), wine (χ^2^ = 11,348, *p* = 0.023), liquor (χ^2^ = 14,603, *p* = 0.006) and soft drinks (χ^2^ = 16,663, *p* = 0.002 ).

Respondents cohabiting with others indicated stronger increases in jogging/running (χ^2^ = 16,381, *p* = 0.003), walking (χ^2^ = 26,473, *p* < 0.001) and cycling (χ^2^ = 10,433, *p* = 0.034). Respondents living alone reported more sedentary behaviour (χ^2^ = 9594, *p* = 0.048).

The other nutritional and PA parameters were not statistically significant (*p* > 0.05).

### 3.6. Changes in Nutritional and PA Parameters According to Family Situation (Living with Others versus Living with Children)

Respondents living with others showed stronger increases in wine (χ^2^ = 19,519, *p* = 0.001) (as opposed to decreases in cohabiting with children) and sweets (χ^2^ = 21,146, *p* < 0.001) (as opposed to decreases in cohabiting with children) consumption. In addition, they were physically more active, due to an increase in the PA parameters of walking (χ^2^=15,575, *p* = 0.004) and jogging/running (χ^2^ = 12,871, *p* = 0.012).

Respondents living with children showed stronger declines in beer (χ^2^=28,846, *p* < 0.001), liquor (χ^2^ = 17,153, *p* = 0.002), soft drinks (χ^2^ = 13,507, *p* = 0.009) and sweets (χ^2^ = 21,146, *p* < 0.001) (as opposed to increases in cohabiting with others) consumption. Moreover, respondents living with children ordered less takeaway/delivery restaurant/brasserie/bistro (χ^2^ = 10,577, *p* = 0.032) and takeaway/delivery meal boxes (χ^2^ = 11,517, *p* = 0.021) and stronger increases in the number of snacks (χ^2^ = 10,266, *p* = 0.036) and sedentary behaviour (χ^2^ = 29,429, *p* < 0.001).

The other nutritional and PA parameters were not statistically significant (*p* > 0.05).

### 3.7. Changes in Nutritional and PA Parameters according to Professional Situation (Students versus the Inactive Population versus the Employed)

Students showed the strongest increases in fresh fruit (χ^2^=61,779, *p* < 0.001) and fresh vegetables (χ^2^ = 22,920, *p* = 0.003) consumption. Morover, pasta (χ^2^ = 22,364, *p* = 0.004), milk and dairy (χ^2^ = 30,322, *p* < 0.001), water (χ^2^ = 102,250, *p* < 0.001), slices of cheese (slices) (χ^2^ = 41,307, *p* < 0.001), cookies, cake and/or pie (χ^2^ = 45,669, *p* < 0.001) and sedentary behaviour (χ^2^ = 130,949, *p* < 0.001) were increased in this population.

The inactive population showed the strongest decrease in plant-based dairy (χ^2^ = 73,263, *p* < 0.001), ice cream and/or sorbet (χ^2^ = 33,914, *p* < 0.001), sauces (χ^2^ = 31,729, *p* < 0.001), jogging/running (χ^2^ = 84,271, *p* < 0.001) and strength training with own body weight (χ^2^ = 119,575, *p* < 0.001) (all in contrast to increases in the others). They also showed the strongest drop in strength training with machines (χ^2^ = 53,367, *p* < 0.001) compared to the other groups.

Employed respondents showed the greatest increases in number of main meals (χ^2^ = 41,994, *p* < 0.001) (as opposed to decreases in the others) and number of snacks (χ^2^ = 48,284, *p* < 0.001). Furthermore, a significant increase in consumption of processed meat products (χ^2^ = 33,840, *p* < 0.001) (as opposed to decreases in the others), chocolate (χ^2^ = 27,909, *p* < 0.001), wine (χ^2^ = 67,016, *p* < 0.001) (as opposed to decreases in the others), sweets (χ^2^ = 42,168, *p* < 0.001) (as opposed to dips in the others), crisps (χ^2^ = 34,354, *p* < 0.001), diet soda (χ^2^ = 37,346, *p* < 0.001) (as opposed to dips in the others), and sweet spreads (χ^2^ = 37,198, *p* < 0.001) (as opposed to dips at the others) was seen. Walking (χ^2^ = 77,239, *p* < 0.001) and cycling (χ^2^ = 39,867, *p* < 0.001) were also significantly increased in this population. They show a less steep drop in liquor (χ^2^ = 58,621, *p* < 0.001) consumption compared to the other groups.

The other nutritional and PA parameters were not statistically significant (*p* > 0.05).

## 4. Discussion

The main goal of the present study was to investigate adjustments in food choices and PA of Flemish adults during lockdown. In general, there were more changes in PA than in food choices, where adjustments in food choices were both positively and negatively linked to health. During lockdown, more time was spent at home, where in principle there are more options to prepare meals. An increase in consumption of fresh foods and a strong reduction in takeout/delivery, found in the present study, confirms this. Our findings corroborate the trend seen by Di Renzo et al. (2020) and Rodriguez-Perez et al. (2020), where during lockdown both Italian and Spanish subjects chose to purchase more fruits and vegetables. The results of the present study also corroborate other studies reporting a decrease in fast-food consumption and a decreased frequency of ordered food during lockdown [15,16,17,19,20,21]. Most studies showed an increase in snacking during lockdown, while in the present study mainly an increase was observed in employed respondents, probably due to stress-related factors and more access to snacks during telework [17,18,24,25,26,27].

Moreover, a distinct link was demonstrated between adjustments in food choices and PA with demographic variables. The middle age group (35–50 year old) showed the most unhealthy adjustments in food choices compared to the other age groups, while the oldest age group (51–81 year old) was strongly affected in their movement pattern in contrast to the other age groups. Gornicka et al. (2020) also found that unhealthy food habits were more present in subjects aged 40 years and older [23]. Acute psychological stress can affect caloric intake, as well as changes in food choices, favouring more foods rich in simple carbohydrates, saturated fats and salt when stress levels are high [39,40,41,42]. Comfort food gives people a sense of emotional well-being, especially to relieve feelings of fear, sadness and other negative emotions [43]. Foods rich in simple carbohydrates or sugars can reduce stress as they enhance serotonin production which has a positive effect on mood [44].

Subjects living in an urban area seemed to adjust their dietary choices in an unhealthy manner, as well as increasing their sedentary behaviour and reducing their cycling activities, whereas subjects living in a suburban area or a rural area showed a significant increase in both cycling and walking activities. Research regarding these data is scarce, so comparing with similar studies was not possible.

In addition, subjects living together with others showed a significant increase, and thus a healthier change, in PA. Food choices of respondents living alone generally were more favourable towards healthy choices than those cohabiting with others. The dietary adjustments of subjects cohabiting with children were generally healthier than those of cohabiting cohousing with other adults. This is in contrast to the study of Gornicka et al. (2020), where unhealthy food habits were more present in respondents living with children [23]. The population of students showed some positive dietary adjustments, but also some negative ones. For many, the positive adjustments are probably due to staying at home again, where healthier cooking and less quick solutions such as ready-made meals or takeout/delivery are chosen. Compared to the employed and the inactive (unemployed respondents/job seekers, people who receive sickness benefits, and respondents in the (pre-)retirement) population, students showed the strongest increase in sedentary behaviour, possibly due to the switch to e-learning. This increase in sedentary behaviour was reported by Stults-Kolehmainen et al. (2014) in times of acute psychological stress [9], although an increase in sedentary behaviour during lockdown is probably due to the shift to telework and e-learning. These findings confirm the decrease in PA and increase in sedentary behaviour as seen in other studies [32,33,34]. The inactive population displayed the least adjustments and were quite stuck with their pre-lockdown food choices. In terms of food choices, employed respondents showed the least favourable evolution towards an unhealthy pattern, presumably due to the great impact on work-life balance.

Furthermore, an important conclusion considering nutritional changes during lockdown is that the common phenomenon of panic buying of storable consumer goods does not equal consumption behaviour. This is evident, as our study shows that non-perishable foods with a long shelf life such as pasta and rice were not consumed more. This is a meaningful result for future studies that use purchasing behaviour to explain consumption behaviour [45].

Regarding PA and sedentary behaviour it is clear that in general Flemish adults increased remarkedly not only their walking and cycling activities, but also their sedentary behaviour. Specifically, the employed population reported an increase in walking and cycling, possibly to cope better with the negative consequences of lockdown [46]. Some subgroups such as the youngest age group (18–34 year old), subjects living in a urban area, respondents living with children and students showed a strong increase in sedentary behaviour. In general, 55.3% of all the subjects reported an increase in sedentary behaviour, which is in line with the findings of other studies where an increase in screen time was found [23,30].

Only a few studies investigated both adjustments in nutritional habits and PA during lockdown, and to our knowledge this is the first Belgian study investigating this. In addition most studies focused on the comparison of dietary habits in general (for example: diet quality, environmental impact of the diet, etc.) before and during lockdown, whereas in the present study the different food groups and types of PA were mapped in detail using a 5-point Likert scale. Our study further contributes to identifying target groups, which impacted their health in a negative way during lockdown. These individuals -in the present study identified as employed respondents, 35–50 year old respondents, respondents living alone, respondents living in an urban area- can benefit even more from the guidance of a dietitian, especially as obesity is linked to a higher hospitalization risk and worse outcome after COVID-19 infection [47]. A limitation of the present study is the use of an online questionnaire, where with-in-sample validity could not be explored as all questions were surveyed once each. Hence, this data collection method is susceptible to social desirability bias, which might have had an impact on the accuracy of the results. However, as people were recommended to stay at home during the lockdown, this seemed the most appropriate method to gather information about changes in food choices and types of PA. A second limitation is the representativity of the sample, as on average healthier subjects tend to participate in health-related studies [48]. Taking this into account, it is striking that even these people show major adjustments in food choices and PA. However our findings emphasize the importance of longitudinal research regarding health and lifestyle choices linked to pandemic issues during lockdown or quarantine, and beyond this period.

## 5. Conclusions

The present study demonstrates that certain people experience a lot of adjustments in food choices and PA during lockdown. In any future lockdowns the ability to maintain healthy food choices and PA habits for the entire population is extremely important, as otherwise there will be an increasingly unhealthy population that is less armed against new diseases. Good physical health, which is mostly a result of healthy nutritional habits and regular PA, is essential for a strong immune system and makes individuals more resilient against stress. Future research should focus on encouraging a healthy lifestyle during lockdown and beyond this period.

## Figures and Tables

**Figure 1 nutrients-13-03794-f001:**
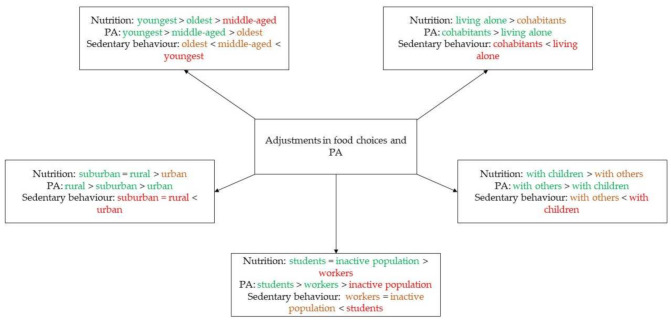
Schematic overview of the results about the relationship between adjustments in food choices and PA with variables such as age, area of residence, home situation, family situation and work situation.

**Table 1 nutrients-13-03794-t001:** Items which were questioned.

Number of main meals
Number of snacks
Portion size of the main meals
Portion size of the snacks
Takeaway/delivery: French fries
Takeaway/delivery: pizza
Takeaway/delivery: food international
Takeaway/delivery: (local) restaurant/brasserie/bistro
Takeaway/delivery: meal boxes
Fresh fruit
Frozen fruit
Canned fruit/fruit in a jar
Freeze-(dried) fruit
Fresh vegetables
Frozen vegetables
Canned vegetables/vegetables in a jar
Fresh potatoes
Ready-made potato preparations
Fried potato preparations
Pasta
Rice
Couscous, bulgur, quinoa, buckwheat, Ebly, etc.
Fresh meat
Canned meat/meat in jar
Fozen meat
Red meat
White meat
Minced meat (preparations)
Processed meat products
Fresh fish
Canned fish/fish in a jar
Frozen fish
Smoked fish
White fish/lean fish
Colored fish/fatty fish
Fresh seafood
Canned seafood/seafood in a jar
Frozen seafood
Eggs
Dried legumes
Canned legumes/legumes in a jar
Seitan, tempeh, tofu
More processed vegetarian/vegan protein sources such as burgers, sausages, etc.
Milk and dairy
Plant-based alternatives to milk and dairy products
Fresh cheese
Slices of cheese
Soft cheese
Cheese spread
Margarine to bake
Margarine to spread
Oils
Ready-made meals from the fresh market and frozen in the supermarket
Candy
Chocolate
Cookies, cake and/or pie
Ice cream and/or sorbet
Crisps
Sweet spreads
Savory spreads
Sauces
Unsalted nuts
Salted nuts
Beer
Wine
Liquor
Soft drinks
Diet soda
Water (including sparkling water and flavoured water)
Coffee and/of tea
Walking
Cycling
Jogging/running
Strength training with own body weight
Strength training with external weights and/or machines
Sedentary behaviour
Sports through (computer) games

**Table 2 nutrients-13-03794-t002:** Descriptive data of the respondents.

	Mean	SD	Minimum	Maximum
Age (years)	34.9	14.3	18.0	81.0
	*n*	%		
Men	253	22.4		
Women	870	77.1		
X	6	0.5		
	*n*	%		
Urban	325	28.8		
Suburban	452	40.0		
Rural	352	31.2		
	*n*	%		
Living alone	163	14.4		
Living together	966	85.6		
	*n*	%		
Living with others	613	54.3		
Living with children	353	31.3		
	*n*	%		
Students	389	34.5		
Inactive population ^£^	118	10.4		
Active population ^§^	622	55.1		
	*n*	%		
Less time preparing meals	20	1.8		
Just as much time preparing meals	362	32.1		
More time preparing meals	344	30.5		

SD = standard deviation; *n* = number of respondents; % = percentage of respondents; ^£^ Unemployed people/job seekers, people who receive sickness benefits, people in (pre-)retirement; ^§^ Employed respondents.

**Table 3 nutrients-13-03794-t003:** Most stable and least stable parameters (in percentages of score 3 (= “I consume/do this now the same as before the lockdown”)).

Most Stable Parameters—Least Changes in Both Directions ^£^	Least Stable Parameters—Most Changes in Both Directions ^§^
Margarine to bake (84.4%)	Number of snacks (45.5%)
Margarine to spread (84.1%)	Takeaway/delivery: (local) restaurant/brasserie/bistro (46.8%)
Oils (83.4%)	Chocolate (44.6%)
Red meat (83.1%)	Cookies. cake and/or pie (48.0%)
Rice (82.3%)	Water (including sparkling water and flavoured water) (49.6%)
White meat (81.9%)	Walking (19.7%)
Cheese spread (81.2%)	Cycling (41.3%)
Number of main meals (81.2%)	Sedentary behaviour (33.0%)
Processed meat products (80.3%)	
Canned seafood/seafood in a jar (80.3%)	

^£^ More than 80% of the respondents reported a score of 3 (= “I consume/do this now the same as before the lockdown”). ^§^ Less than 50% of the respondents reported a score of 3 (= “I consume/do this now the same as before the lockdown”).

**Table 4 nutrients-13-03794-t004:** Strongest risers and fallers in percentages.

Strongest Risers ^£^	Strongest Fallers ^§^
36.5% cookies, cake and/or pie	46.0% takeaway/delivery: food international
37.1% coffee and/or tea	44.4% takeaway/delivery: French fries
37.9% cycling	43.4% takeaway/delivery: (local) restaurant/brasserie/bistro
39.2% fresh vegetables	42.8% takeaway/delivery: pizza
40.3% eggs	
40.3% chocolate	
42.4% fresh fruit	
44.8% water (including sparkling water and flavoured water)	
55.3% sedentary behaviour	
62.7% walking	

^£^ Percentage of respondents who reported to consume/do this now (much) more than before the lockdown. ^§^ Percentage of respondents who reported to consume/do this now (much) less than before the lockdown.

## Data Availability

The data presented in this study are available in Table 1, Table 2, Table 3 and Table 4 and Appendix A.

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
