# Peer review of "Adjustments in Food Choices and Physical Activity during Lockdown by Flemish Adults"

_nutrients, 2021, doi:10.3390/nu13113794_

Round 1

Reviewer 1 Report

Dear authors, 

This review is quite well-written and informative. 

Please consider existing systematic-reviews such as 

1) Zaccagni L, Toselli S, Barbieri D. Physical Activity during COVID-19 Lockdown in Italy: A Systematic Review. Int J Environ Res Public Health. 2021 Jun 13;18(12):6416. doi: 10.3390/ijerph18126416. PMID: 34199286; PMCID: PMC8296244. 

2)  Bakaloudi DR, Jeyakumar DT, Jayawardena R, Chourdakis M. The impact of COVID-19 lockdown on snacking habits, fast-food and alcohol consumption: A systematic review of the evidence. Clin Nutr. 2021 Apr 17:S0261-5614(21)00212-0. doi: 10.1016/j.clnu.2021.04.020. Epub ahead of print. PMID: 34049747; PMCID: PMC8052604.

  Which provide information regarding PA and eating habits during the lockdown period.   Moreover, the fact that data collection was performed by online questionnaires could have an impact on the accuracy of the results and this should be added in the limitation section.

Reviewer 2 Report

Dear authors, I have some recommendations about your paper:

  1. in lines 67-69 you wrote “So 67 far, data on dietary changes during lockdown are scarce and most studies focused on diet 68 quality and/or quantity during lockdown” and nowadays a lot paper about change in dietary habits in people during lockdown all around the world. I recommended the authors to include more references about that in the Introduction.
  2. Table 1. should have the same type of letter than the main text
  3. I recommended the authors to improve the presentation and quality of data in Table 2 and change the type of letter
  4. Table 3 and Table 4 explanations are not very clear. Change the type of letter. I recommended the authors to explain it better.
  5. from line 199 to 282 I recommended the authors to do some Tables to facilitate the understanding of all this data

Round 2

Reviewer 2 Report

Thank you for improving your paper
